# Immediate Implant Placement with Soft Tissue Augmentation Using Acellular Dermal Matrix Versus Connective Tissue Graft: A Systematic Review and Meta-Analysis

**DOI:** 10.3390/ma17215285

**Published:** 2024-10-30

**Authors:** Andrea Galve-Huertas, Louis Decadt, Susana García-González, Federico Hernández-Alfaro, Samir Aboul-Hosn Centenero

**Affiliations:** Department of Oral and Maxillofacial Surgery, Universitat Internacional de Catalunya, 08195 Sant Cugat del Vallès, Spain; louis_decadt@uic.es (L.D.); susanagarcia@uic.es (S.G.-G.); samir@uic.es (S.A.-H.C.)

**Keywords:** immediate implant, soft tissue augmentation, acellular dermal matrix, connective tissue, single tooth, soft tissue thickness

## Abstract

This systematic review investigates the efficacy of using connective tissue grafting (CTG) versus an acellular dermal matrix (ADM) for soft tissue management in immediate implant placement (IIP). The study focuses on comparing the soft tissue thickness (STT) and keratinized tissue width (KTW) changes post-implantation. Adhering to the PRISMA guidelines, a comprehensive literature search was conducted, targeting randomized clinical trials and cohort studies involving soft tissue grafting in conjunction with IIP. Data extraction and analysis focused on STT and KTW measurements from baseline to follow-up intervals of at least 6 months. The statistical analyses included the weighted mean differences and heterogeneity assessments among the studies. The meta-analysis revealed no significant difference in the STT gain between CTG and ADM at 12 months, with the weighted mean differences favoring the control group but lacking statistical significance (CTG: 0.46 ± 0.53 mm, *p* = 0.338; ADM: 0.33 ± 0.44 mm, *p* = 0.459). The heterogeneity was high among the studies, with discrepancies notably influenced by individual study variations. Similarly, the changes in KTW were not significantly different between the two grafting materials. Conclusions: Both CTG and ADM are viable options for soft tissue management in IIP, with no significant difference in efficacy regarding the soft tissue thickness and keratinized tissue width outcomes. Future research should aim to minimize the heterogeneity and explore the long-term effects to better inform clinical decisions.

## 1. Introduction

Achieving optimal aesthetic outcomes in implant dentistry, particularly in the anterior region, remains a paramount goal for both clinicians and patients. Immediate implant placement (IIP) following tooth extraction has gained considerable attention due to its potential advantages in reducing the treatment time and preserving the peri-implant tissue [1,2,3,4].

Obtaining successful outcomes in the aesthetic zone demands meticulous planning, precise execution by clinicians with clinical experience and expertise, and the strategic management of soft tissue contours and biotypes. The restoration of missing teeth in the anterior maxilla demands not only osseointegration, but also the establishment of harmonious gingival contours that mimic natural dentition. Achieving and maintaining a stable soft tissue architecture around the implant is pivotal in creating a lifelike appearance and ensuring long-term aesthetic success [1,2,3,4].

A tooth extraction results in a series of changes occurring in the alveolar bone, which includes a bone remodeling phase. In the initial phase, due to the loss of the periodontal ligament and loss of the bundle bone, the lingual and buccal plate post-extraction undergo a series of changes that includes horizontal and vertical changes. In the anterior sector, this can compromise the aesthetics of the patient, especially if the extraction procedure was not atraumatic [5,6].

As a result of the volumetric changes, management to avoid unfavorable results has to be carried out [6].

Funato et al. describes and classifies the four scenarios in which immediate implants can be considered. However, he explains that extraction sockets with four walls show better aesthetic predictability [7]. On the other hand, Buser et al. describes protocols according to the time of placement, with the type 1 protocol involving immediate implant placement (IIP) directly into the alveolar socket via flapless implant placement with osseous augmentation and a connective tissue graft (CTG) or allograft, being a predictable approach [8].

Following the work of Chu et al., the classification of the different types of sockets has been used as a tool to aid clinicians in carrying out surgical interventions with several protocols known to fill the gap [9].

It is not only the management of hard tissue that must be considered in immediate implant therapy, but also that of soft tissue. This is due to the risk of deterioration in the soft tissue in the follow-up period; horizontal ridge deficiency, which results in the loss of the mid-facial contour; soft tissue recessions; and shine-through discoloration, leading to aesthetic compromises [10,11,12].

Considering the series of changes following a tooth extraction when placing an immediate implant, the patient characteristics also need to be evaluated, including the buccal position of the implant, the hard tissue status according to the classification of Chu et al., and the patient’s phenotype [9,11].

Soft tissue was previously left untreated. Various studies have described results favoring the use of an autologous connective tissue graft (aCTG) when compared with the use of no soft tissue grafting on the peri-implant tissue’s health and stability, due to decreasing the risk and prevalence of peri-implant mucositis and peri-implantitis by reducing the horizontal changes in the alveolar ridge. Moreover, it allows the maintenance of the tissue contour due to an increase in the soft tissue thickness, leading to a better aesthetic result from both the patient and clinician’s point of view [13,14].

Soft tissue augmentation is recommended and has been established as part of the gold-standard protocol for immediate implant therapy, particularly for patients with a thin periodontal phenotype [15]. This phenotype is typically associated with a thin buccal bone, which undergoes greater bone resorption and soft tissue contraction in the postoperative phase [9,11]. However, autologous CTG sources are associated with disadvantages, such as postoperative pain and increased medication use, which can negatively affect patient-reported outcome measures (PROMs) [16,17].

Tavelli et al. found that various soft tissue grafting techniques, including free gingival grafts and connective tissue grafts, significantly increase the mucosal thickness around dental implants compared to sites without grafts, leading to improved peri-implant health and stability [18].

The use of an acellular dermal matrix (ADM) as an alternative method for soft tissue grafting has become more favorable, with the benefits not only favoring the clinician but also improving the PROMs in the postoperative stage following the surgical intervention [16,17]. In the case of a lack of sufficient donor material, the ADM has been developed as a substitute for the aCTG and has presented a potential alternative to thicken the peri-implant soft tissue. An ADM is a type of biomaterial derived from human or animal tissue that has been processed to remove cells, leaving behind the extracellular matrix (ECM). The ECM is composed of proteins such as collagen, elastin, and glycosaminoglycans, which provide structural support and signaling cues for tissue regeneration.

Following a study conducted by Stefanini et al. on soft tissue augmentation around implants, it was concluded that surgical sites undergoing soft tissue augmentation typically retain their soft tissue margins and marginal bone levels over time, whereas implants without augmentation may experience the recession of the soft tissue margin [19]. This study found that the clinical parameters remained stable over time for both CTG and ADM grafts. These findings are consistent with the results of Tommasato et al., who also reported increased an mucosal thickness postoperatively with ADM grafts [20].

One of the key advantages of an ADM is its ability to be integrated with the patient’s own tissue, promoting natural healing and minimizing the risk of rejection [16,17,18,19,20].

For these reasons, the aim of this article is to investigate the critical aspects of IIP, focusing on the challenges that clinicians face in the management of soft tissue volumetric changes during immediate implant therapy, as well as looking for any changes in keratinized tissue when the two soft tissue substitutes are compared. Therefore, the objective of the present systematic review is to compare the effectiveness of CTG versus ADM for soft tissue management in IIP.

## 2. Materials and Methods

### 2.1. Clinical Question and Search Strategy

This systematic review was designed in accordance with the Preferred Reporting Items for Systematic Review and Meta-Analyses (PRISMA) 2020 statement (Appendix A) [18], using the following population, intervention, comparison, outcome, study design (PICOS) model.

Population (P): Healthy patients receiving an immediate implant due to the tooth being unrestorable with a simultaneous soft tissue graft in all sites of the mouth.Intervention (I): Soft tissue grafting with an ADM in conjunction with IIP.Comparison (C): Soft tissue grafting with an aCTG in conjunction with IIP; differences in soft tissue thickness and keratinized tissue changes under IIP with either an ADM or an aCTG.Outcome (O): The soft tissue thickness and keratinized tissue changes when comparing the different soft tissue graft materials from baseline, during the surgical act of placing the immediate implant, to the follow-up interval of a minimum of 6 months, using an analog or digital volumetric analysis.Study Design (S): To evaluate human randomized clinical trials (RCTs), with prospective/retrospective cohorts, that evaluate the peri-implant tissue following soft tissue augmentation with the different grafts.

The main question was, “In systemically healthy patients, being treated by means of single IIP with a simultaneous soft tissue augmentation procedure, is the soft tissue thickness and the keratinized tissue width affected by the type of soft tissue graft used when comparing an acellular dermal matrix graft to aCTG?”

Electronic and manual literature searches were conducted by two independent reviewers (LD and AG) in the National Library of Medicine (Medline via PubMed) [USA] for articles published up until March 2024.

The following search terms were used: ((“immediate”[All Fields] OR “immediately”[All Fields]) AND (“dental implants”[MeSH Terms] OR (“dental”[All Fields] AND “implants”[All Fields]) OR “dental implants”[All Fields]) AND (“soft”[All Fields] AND (“tissue s”[All Fields] OR “tissues”[MeSH Terms] OR “tissues”[All Fields] OR “tissue”[All Fields]) AND (“augment”[All Fields] OR “augmentation”[All Fields] OR “augmentations”[All Fields] OR “augmented”[All Fields] OR “augmenting”[All Fields] OR “augments”[All Fields]))) NOT (“embryo s”[All Fields] OR “embryoes”[All Fields] OR “embryonic structures”[MeSH Terms] OR (“embryonic”[All Fields] AND “structures”[All Fields]) OR “embryonic structures”[All Fields] OR “embryo”[All Fields] OR “embryos”[All Fields]).

The study was prospectively registered in the International Prospective Register of Systematic Reviews (PROSPERO) database (ID: CRD42024571855). Moreover, a flowchart illustrating the search strategy and selection process was created (Figure 1).

### 2.2. Eligibility Criteria

The inclusion criteria were as follows: (1) publications in English; (2) human studies; (3) RCTs and prospective and retrospective cohort or case series studies where there were two groups to allow comparison; (4) the variables in the studies compared at least two groups of IIP + ADM versus IIP with aCTG; (5) healthy patients received an immediate implant with a soft tissue graft (CTG or ADM); (6) the studies had a minimum follow-up period of 6 months; (7) the reported outcome variables of the studies included the soft tissue thickness, the keratinized tissue width, and its standard deviation from the baseline to the follow-up interval.

Exclusion criteria: (1) studies that did not describe the outcome variables of the soft tissue thickness or keratinized tissue width; (2) studies that did not compare human ADMs versus aCTG as soft tissue substitutes in soft tissue augmentation in an immediate implant protocol; (3) publications in languages other than English, case reports, educational statements, expert opinions, animal studies, in vitro studies, narrative reviews on the subject of soft tissue grafting on immediate implants, or studies without a control group; (4) studies that did not provide any information concerning the research question.

### 2.3. Selection Process

In the first phase of the study selection, two reviewers (LD and AG) screened all identified titles and abstracts independently to assess their eligibility for the systematic review based on the predetermined inclusion and exclusion criteria.

In the second phase of study selection, the full-text articles of all studies identified in the first phase were retrieved and evaluated by the authors (LD and AG) on an independent basis. Any disagreements between the authors were resolved by discussion with two additional reviewers (SG and SAH).

### 2.4. Data Analysis

Full-text data extraction was performed independently for each eligible article by two reviewers (LD and AG).

The following variables were extracted from each study: (1) author(s); (2) year of publication; (3) type of study; (4) follow-up (FU); (5) study outcomes (soft tissue thickness, keratinized mucosa width, VAS, gingival biotype, pocket depth, peri-implant mucosal level, marginal bone level, bleeding index scores, perceived pain, VAS, and PES) (Table 1).

Additionally, the surgical protocol of each study was summarized and they were divided into a test group (ADM group) and control group (CTG): (1) author and year; (2) follow-up (FU); (3) number of implants; (4) methodology; (5) area of graft (palate or tuberosity); (6) flap or flawless design; (7) use of biomaterial or not; (8) gingival phenotype reported.

### 2.5. Risk of Bias and Quality of the Studies

Two reviewers (LD and AG) designed and assessed the proposal for the present project to ensure that the [21] guidelines were followed (Found in Appendix A), as well as the use of the Strengthening the Reporting of Observational Studies in Epidemiology (STROBE) [22] statement. The STROBE statement is an international collaborative initiative for epidemiologists, methodologists, statisticians, researchers, and journal editors involved in the conduction and dissemination of observational studies and consists of a 22-item checklist to be fulfilled in a systematic review.

The quality of the included RCTs was established from the RCT checklist of the Cochrane Center and the Consolidated Standards of Reporting Trials (CONSORT) statement, which provided guidelines for the following parameters [23,24]: sequence generation; allocation concealment method; masking the examiner; addressing incomplete outcome data; free of selective outcome reporting.

### 2.6. Outcome Analysis

The primary outcome of the systematic review was the change in soft tissue thickness (STT) determined at the baseline and at the final follow-up, evaluated by comparing the volume through either an analogical method using manual calipers or K-files or by comparing digital STL files following an impression at both the baseline and follow-up periods.

The secondary outcome was the change in the keratinized mucosal width, evaluated using a periodontal probe, measured from the mucogingival junction towards the free gingival margin.

**Table 1 materials-17-05285-t001:** Characteristics of included studies and methodologies used to obtain STT and KTW outcomes.

	Study Design	Nº of Patients	Nº of Implants	Tooth Replacement	Groups	Immediate Provisionalization	Follow-Up	Method of Measurement of Outcomes for STT and KTW
Panwar et al., 2021 [25]	Randomized Controlled Trial	20	20	Superior anterior zone (from second premolar to second premolar)	Group 1: IIP + ADM Group 2: IIP + CTG	None	6 m	1. Soft tissue thickness: use of Vernier calipers. 2. Keratinized mucosa width changes: use of a periodontal probe (measured from muco-gingival junction to mid-buccal gingival or per-implant mucosal level).
Happe et al., 2021 [26]	Randomized Controlled Trial	20	20	Superior anterior zone (from second premolar to second premolar)	Group 1: IIP + ADM Group 2: IIP + CTG	Not reported	12 m	1. Soft tissue thickness: superimposition of Standard Tessellation Language (STL) files in a digital imaging program.
Lee et al., 2023 [27]	Randomized Controlled Trial	30	30	Superior anterior zone (from second premolar to second premolar)	Group 1: IIP + ADM Group 2: IIP + CTG Group 3: IIP without graft	Healing abutment	12 m	1. Soft tissue thickness: use of endodontic files and use of digital impressions. 2. Keratinized mucosa width changes: use of a periodontal probe (measured from muco-gingival junction to mid-buccal gingival or per-implant mucosal level).
Abbas et al., 2020 [28]	Randomized Controlled Trial	14	14	Superior and inferior zone	Group 1: IIP + ADM Group 2: IIP + CTG	Temporary prothesis	12 m	1. KTW: use of a periodontal probe (measured from muco-gingival junction to mid-buccal gingival or peri-implant mucosal level).
De Angelis et al., 2023 [29]	Prospective Clinical Trial	48	48	Superior anterior zone (from second premolar to second premolar)	Group 1: IIP + ADM Group 2: IIP + CTG	Depending on stability; 1 or 2 stages followed by a provisional.	12 m	1. Soft tissue thickness: use of endodontic files and use of digital impressions.

IIP: Immediate Implant Placement, CTG: Connective Tissue Graft, ADM: Acellular Dermal Matrix.

### 2.7. Statistical Analysis

A meta-analysis was carried out to estimate the global effect size (ΔSTT Baseline-12mo) in each of the groups. This was followed by a comparative intra-study meta-analysis between the two types of material, with the ADM being compared directly to CTG. Weighted mean differences (WMD) and 95% confidence intervals were calculated from random effects models with the DerSimonian and Laird estimator.

The results of the estimates, global effect measures, and confidence intervals are represented in the different forest graphs.

The heterogeneity index I^2^ (percentage of variability in the estimated effect that could be attributed to heterogeneity of the true effects) and the corresponding statistical test of nullity of Q were applied. A threshold of I^2^ < 50% was considered as an acceptable level of between-study heterogeneity. A forest plot was produced for each outcome to represent the difference graphically.

Publication bias was explored through funnel plots and the Egger test.

The level of significance used in the analyses was 5% (α = 0.05).

The software used to carry out the meta-analysis was R 4.3.1 (R Core Team (2023), R: A language and environment for statistical computing, R Foundation for Statistical Computing, Vienna, Austria.

## 3. Results

### 3.1. Study Selection

The electronic search rendered 244 titles in total (Figure 1). One potentially eligible article was identified on the basis of a manual search. The gray literature did not result in any additional articles. Eleven articles of the remaining 76 were excluded after full-text analysis. The remaining eight articles were assessed for eligibility. Ultimately, five articles fully met the selection criteria for a qualitative analysis [25,26,27,28,29], as described in Table 1.

Exclusion was due to either the methodology or the primary outcome investigated. The articles of Guglielmi et al. [13] and Zuiderveld et al. [30] were excluded due to comparisons of soft tissue augmentation with an aCTG compared to no graft. Additionally, articles [31,32] were excluded due to the outcome variable not reflecting the outcome variable of this systematic review.

### 3.2. Intervention Types and Sample Characteristics

Table 1 illustrates the studies included. Four were RCT studies and one was a prospective study. All five studies were two-arm types comparing a test group (ADM) and control group (CTG) with IIP. The study of Lee et al. had three groups: test groups of IIP + CTG and IIP + ADM with a control of IIP placed with no graft (NG). Regarding the NG group, the data were excluded from this systematic review [27].

The total number of implants involved in the studies was 132: 66 in ADMs and 66 in CTG.

The main outcomes of the studies were the variations in the STT and KTW in both groups, with follow-up periods of a minimum of 6 months. Only some studies reported the mean variation with the SD, which was included in the meta-analysis [25,27,29]. For the meta-analysis, only three had 12-month information for the STT [25,27,29] and only one for the KTW [27]. Therefore, the statistical analysis was restricted to the variation in the STT at a follow-up period of 12 months, between the ADM group and the aCTG control group.

All surgeries were performed under local anesthesia, with the majority carrying out the surgery with a flapless approach, apart from Panwar et al. [25], where a non-traumatic extraction was carried out, followed by the debridement of the alveolar socket. All studies that were included in the systematic review used bone-level implants, leaving a buccal gap from the bone wall. Panwar et al. did not fill the gap with a biomaterial [25]. Three studies reported filling the gap with a xenograft [26,27,29]. Following this, the studies reported the division of two groups where an ADM graft was used and compared these to an aCTG obtained from the palate. Both grafts in all studies were sutured with sling sutures.

All studies had different rehabilitation procedures, with two studies not reporting provisional restoration rehabilitation [26,27]; however, all implants were reported as restored at 3 to 6 months postoperatively.

### 3.3. Soft Tissue Thickness Changes for CTG Group at 12 Months

The appropriate and valid studies for this analysis are described in Table 2.

The STT gain estimate is 0.46 ± 0.53 mm, which not considered significant (*p* = 0.388) (Table 3).

The forest plot also provides the relative weight of each study in the meta-analysis estimates. This illustrates that the three studies contributed to the calculations in a similar proportion (Figure 2).

The heterogeneity of the meta-analysis was found to be significant (I^2^ = 98.9%). The explanation for this lies in the great discrepancy between the results of Happe et al. [26] and those of the other two studies [31,33], whose confidence intervals did not intersect each other and hence the inter-study variability was, in general, much higher than the intra-study variability.

On the other hand, publication bias was detected (*p* = 0.013). This result was not conclusive due to the small number of articles, but the funnel graph shows asymmetry (Figure 3).

Due to the small number of implants and higher level of variability, Happe et al.’s study demonstrated a higher standard of error. When compared to the other two articles, the results appear more precise and exhibit the opposite correlation, reporting the true value of the soft tissue thickness.

### 3.4. Soft Tissue Thickness for ADM Group at 12 Months

Table 4 shows the appropriate and valid studies for this analysis.

The forest plot (Figure 4) demonstrates an estimation of the results of the meta-analysis conducted for the soft tissue thickness in the ADM group at 12 months.

The STT gain estimate for the ADM group was 0.33 ± 0.44 mm, and this value was also found not to be significant (*p* = 0.459) (Table 5).

From the forest plot, it can be seen that each article contributed in a similar proportion (Figure 4).

There was a high level of heterogeneity among the values (I^2^ = 98.7%). Following the trend in the other analysis, there was no correlation regarding the results between Happe et al. [26] and the other studies. In this part of the analysis, publication bias could not be detected. This was due to the three studies having a similar level of precision in regard to their analyses (Figure 5).

### 3.5. Soft Tissue Thickness Changes: ADM + IIP vs. CTG + IIP Intra-Study Comparison at 12 Months

The meta-analysis yielded a weighted mean difference (WMD) of −0.14 mm, favoring the control aCTG group, but with no significant statistical difference between the two groups (*p* = 0.490) (Table 6).

The heterogeneity among the results was found to be high, as Happe et al. and Lee et al. obtained similar results when comparing both the CTG and ADM groups. However, this was not the case for De Angelis et al., who reported a greater gain in the soft tissue thickness with CTG (Figure 6).

There was also a small amount of publication bias found among the studies included (*p* = 0.490) (Figure 7).

For this reason, when comparing the use of the ADM and CTG when augmenting the soft tissue around the IIP, the soft tissue changes at 12 months showed similar average gains and no statistical difference.

### 3.6. Keratinized Tissue Width Changes

The results regarding the keratinized width changes reported in the selected studies are shown in Table 7.

The main findings were found in three of the five studies selected in the systematic review, with the results showing heterogeneity regarding when the keratinized tissue changes were reported [25,27,28].

Panwar et al. reported that the CTG group showed a statistically significant increase of 0.65 ± 0.0411 mm in the KTW; however, the gain was smaller than that found in the present literature. The reason for the reported decrease in the KTW in the ADM group was the use of a buccal flap advancement. The reason for this was to avoid the necrosis of the ADM graft due to having avascular properties compared to an aCTG [25].

Lee et al. [27] reported that, at the 12-month visit, the mean keratinized tissue widths were slightly decreased in the groups compared to the baseline. It was reported that the changes in the keratinized tissue width were not significantly different among the groups.

Abbas et al. [28] reported that there was no statistically significant difference found in the keratinized mucosa (KM) between the baseline, 4-month, and 12-month follow-up intervals in either Group 1 (ADM) or Group 2 (aCTG), with *p*-values of 0.338 and 0.156, respectively. Group 1 experienced a loss of 0.858 ± 1.199 mm in the KM, while Group 2 lost an average of 0.357 ± 1.488 mm. At the baseline, 4-month, and 12-month follow-up intervals, there was no statistically significant difference between the two groups, with *p*-values of 0.628, 0.598, and 0.220, respectively.

## 4. Discussion

Based on the literature analysis, there is limited published evidence comparing soft tissue augmentation between ADMs and aCTG in IIP. While there is abundant literature comparing the use of aCTG and ADMs independently or comparatively, especially around natural teeth, studies exploring techniques to enhance the gingival thickness around immediate implants are currently lacking.

The few clinical studies that do compare the two soft tissue graft options primarily evaluate the soft tissue thickness and keratinized tissue changes, while very few assess the PROMs or the PES score [25,26,27,28,29]. The studies by De Angelis et al., Aldhohrah et al., and Seyssens et al. demonstrate that using aCTG in IIP is more advantageous than not using a graft, improving the outcomes in terms of peri-implant marginal recession and marginal bone loss and increasing the soft tissue thickness [18,33,34]. To date, no systematic reviews have been published that solely assess the two soft tissue grafting options in the context of soft tissue augmentation during immediate implant therapy.

In this systematic review, five articles met the final inclusion criteria and were analyzed. Although there were limited data comparing the same primary outcomes, the studies chosen for the meta-analysis had a homogeneous methodology regarding their outcome variables, allowing for a quantitative analysis to be carried out evaluating the soft tissue thickness after 12 months [26,27,29]. A favorable aspect is that the studies involved had similar intervals and methodologies for the evaluation of the outcome variables, resulting in great homogeneity, which made the results highly comparable.

Regarding the soft tissue thickness, all studies investigated this outcome except for one [28]. The results of the meta-analysis showed that the STT for CTG + IIP was 0.46 ± 0.53 mm, and it was 0.33 ± 0.44 mm for ADM + IIP, indicating no statistical differences between the two groups reported across the meta-analysis [26,27,29]. The results of the meta-analysis demonstrated strong heterogeneity. Firstly, Happe et al. [26] reported results for each group that were significantly different from those of the other two articles. On the other hand, De Angelis et al. [29] were the only authors who reported the clear superiority of CTG compared to the other articles, which indicated greater homogeneity.

Concerning this outcome variable, great heterogeneity was observed between the studies, and the risk of bias was higher. This could be associated with the different follow-up intervals among the studies, as well as the varying methodological approaches to evaluating the soft tissue thickness and changes in keratinized tissue. Although all studies used either a manual caliper, endodontic files, or the superimposition of STLs using different types of digital software (Table 1), the heterogeneity in the methodologies regarding the outcome variable means that the results of each study should be interpreted with caution.

Additionally, the changes in the keratinized tissue width were evaluated in three studies. This outcome was considered as the secondary outcome [25,27,29].

Panwar et al. reported that there were no significant differences in the results when comparing thin and thick phenotype groups regarding keratinized mucosal changes. In the surgical protocol, a flap was raised to avoid exposing the ADM graft, which consequently led to a decrease in the amount of keratinized mucosa in the ADM + IIP group. Lee et al. reported that, in a flapless intervention, both groups resulted in a decrease in keratinized mucosa. It was noted that the ADM grafts were exposed during the healing phase or due to the submerged implant position. In contrast, Abbas et al. concluded that, when the phenotype was divided into thin and thick groups, the analysis of the two different grafts showed that they did not influence the width of the peri-implant keratinized tissue. No significant differences from the baseline to the 4- and 12-month follow-up intervals were reported. It was suggested that the patient’s phenotype could affect the final amount of keratinized tissue, as well as raising the question of whether a larger graft size could influence the final result.

It has been reported that the use of a flapless technique shows a positive correlation with the keratinized gingiva thickness [35]; however, the act of raising a flap allows for more accurate implant placement [36]. Nevertheless, Grassi et al. reported that flap elevation does not seem to enhance the risk of increased bone levels, and there is a trend towards better results when using a flapless approach for peri-implant soft tissue [37].

Similarly, Park et al. [38] indicated that ADM grafts around implants in a delayed approach are a good alternative to increase the keratinized mucosa around dental implants. However, due to the lack of vascularity around dental implants, it is crucial for the ADM graft not to be exposed to allow for the integration of the graft within the peri-implant soft tissue following IIP [25]. Both Panwar et al. and Lee et al. experienced the exposure of the ADM grafts; Panwar et al. performed a flap to prevent the exposure and the consequent loss of keratinized tissue, while Lee et al. reported exposure during follow-ups, leading to necrosis and the loss of the grafts.

The results of this systematic review regarding changes in the keratinized width should be interpreted with caution due to the heterogeneity of the surgical protocols in each study, which has been shown to influence the results.

The overall success rate of the implants in this review was reported to be 100%, with minor complications noted in one of the studies [27]. The exposure of the grafts was also a common issue reported in these studies, which affected the final keratinized mucosa results.

The limitations of the present systematic review include the small number of studies that met the eligibility criteria, as well as the limited number of studies comparing the acellular dermal matrix as a substitute directly to an aCTG when performing soft tissue augmentation. Another limitation is the lack of a homogeneous methodology in the clinical studies when evaluating the outcome variables. Nevertheless, it was possible to conduct a thorough analysis of the most important outcomes of these five studies.

## 5. Conclusions

This review found that soft tissue augmentation using ADM with immediate implant placement (IIP) provides comparable results to connective tissue grafts (CTG) from the palate regarding soft tissue thickness and keratinized tissue changes.

Soft Tissue Thickness: ADM with IIP yields outcomes similar to CTG.Keratinized Tissue: Comparable changes observed with ADM and CTG.Research Needs: Additional RCTs and extended follow-ups are required to confirm long-term stability.

ADM appears viable as an alternative, though further studies are recommended.

## Figures and Tables

**Figure 1 materials-17-05285-f001:**
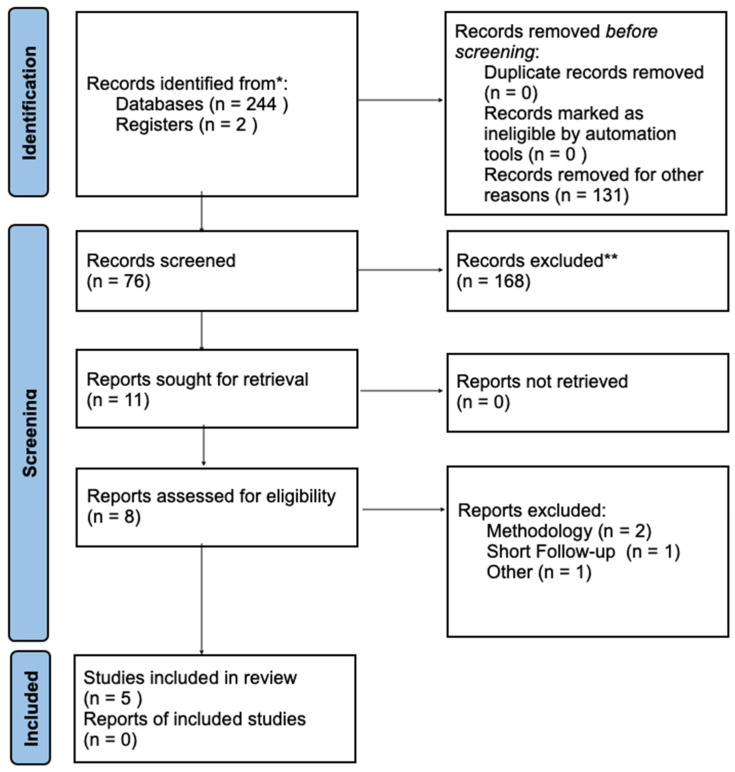
PRISMA flowchart of the search strategy. * Total amount of record identified. ** From the total amount of records identified, 168 Records were excluded after reviewing the abstacts of the studies found in the identification step.

**Figure 2 materials-17-05285-f002:**
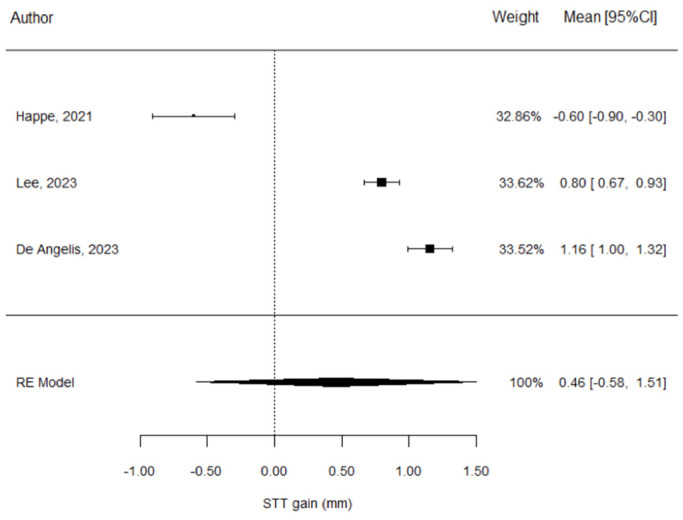
Forest plot for STT in the CTG group; Happe et al. [26], Lee et al. [27], De Angelis et al. [29].

**Figure 3 materials-17-05285-f003:**
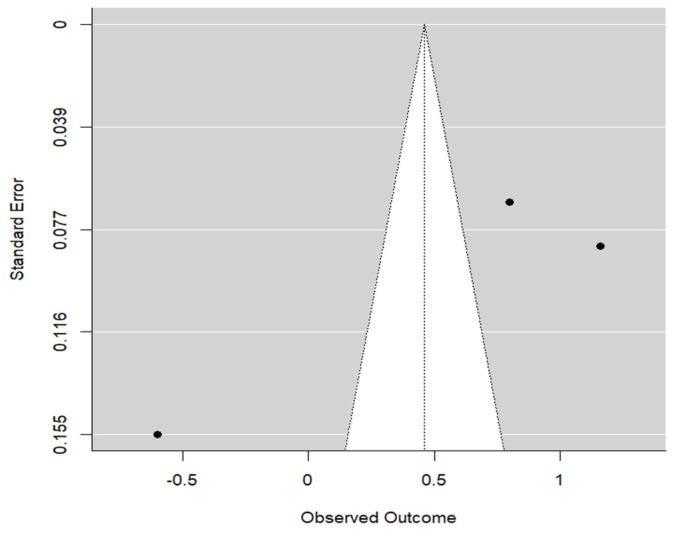
Funnel graph for the STT in the CTG group.

**Figure 4 materials-17-05285-f004:**
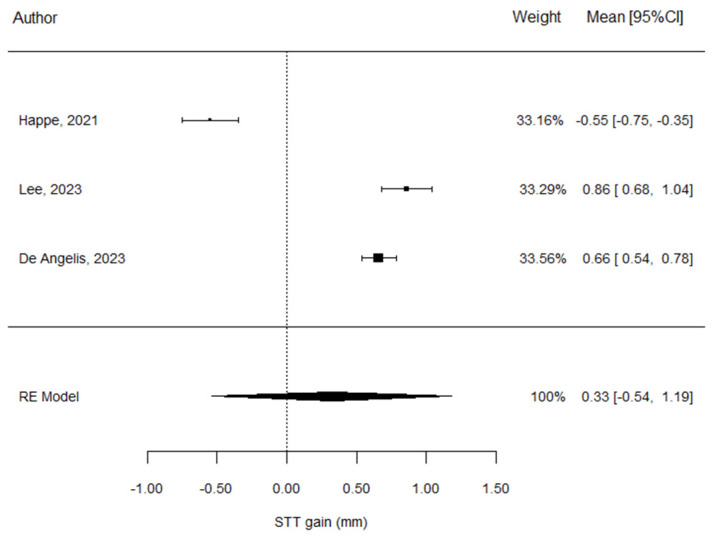
Forest plot for STT in the ADM group; Happe et al. [26], Lee et al. [27], De Angelis et al. [29].

**Figure 5 materials-17-05285-f005:**
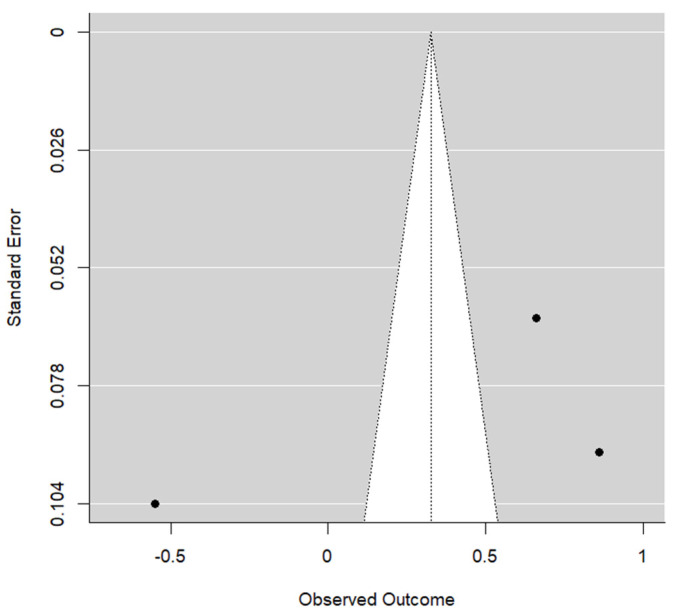
Funnel graph for the STT in the ADM group.

**Figure 6 materials-17-05285-f006:**
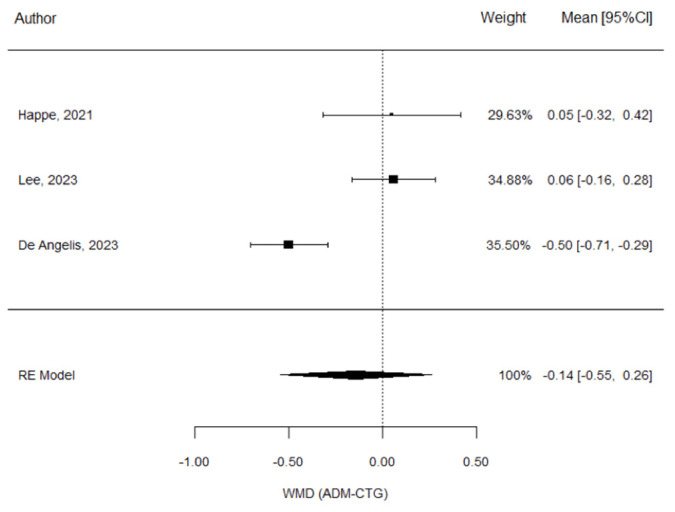
Forest plot for soft tissue thickness changes in ADM + IIP vs. CTG + IIP intra-study comparison at 12 months; Happe et al. [26], Lee et al. [27], De Angelis et al. [29].

**Figure 7 materials-17-05285-f007:**
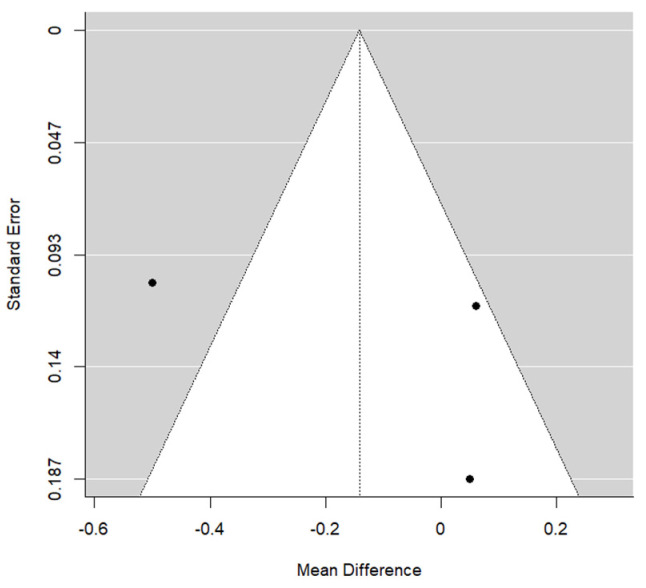
Funnel graph for the soft tissue thickness changes in the ADM + IIP vs. CTG + IIP intra-study comparison at 12 months.

**Table 2 materials-17-05285-t002:** Studies involved in statistical analysis for soft tissue thickness in CTG at 12 months.

	n	Mean	SD
Happe et al., 2021 [26]	10	−0.6	0.49
Lee et al., 2023 [27]	15	0.8	0.26
De Angelis et al., 2023 [29]	24	1.16	0.41

Number (n); standard deviation (SD).

**Table 3 materials-17-05285-t003:** Meta-analysis results for STT in ADM group.

WM	SE	95% CI	Z (*p*-Value)	I^2^	Q_H_ (*p*-Value)	Egger Test (*p*-Value)
0.46	0.53	−0.58 1.51	0.388	98.9%	<0.001 ***	0.013 *

WM: weighted mean; SE: standard error; CI: confidence interval; Z: Z-Test; QH: Cochrane’s Q of heterogeneity. * *p* < 0.05; *** *p* < 0.001.

**Table 4 materials-17-05285-t004:** Studies involved in statistical analysis for soft tissue thickness in ADM at 12 months.

	n	Mean	SD
Happe et al., 2021 [26]	10	−0.55	0.33
Lee et al., 2023 [27]	15	0.86	0.36
De Angelis et al., 2023 [29]	24	0.66	0.31

Number (n); standard deviation (SD).

**Table 5 materials-17-05285-t005:** Meta-analysis results for STT in CTG group.

WM	SE	95% CI	Z (*p*-Value)	I^2^	Q_H_ (*p*-Value)	Egger Test (*p*-Value)
0.33	0.44	−0.54 1.19	0.459	98.7%	<0.001 ***	0.434

WM: weighted mean; SE: standard error; CI: confidence interval; Z: Z-Test; QH: Cochrane’s Q of heterogeneity. *** *p* < 0.001.

**Table 6 materials-17-05285-t006:** Meta-analysis results for the mean differences in the STT gain according to the groups (ADM vs. CTG).

WM	SE	95% CI	Z (*p*-Value)	I^2^	Q_H_ (*p*-Value)	Egger Test (*p*-Value)
−0.14	0.21	−0.55 0.26	0.493	86.8%	<0.001 ***	0.490

WM: weighted mean; SE: standard error; CI: confidence interval; Z: Z-Test; QH: Cochrane’s Q of heterogeneity. *** *p* < 0.001.

**Table 7 materials-17-05285-t007:** Selected studies evaluating keratinized tissue width changes.

		KMW Baseline (Mean ± SD)	KMW 6 m (Mean ± SD)	KMW 12 m (Mean ± SD)	KMW Difference at FU 6 m (Mean ± SD)	KMW Difference at FU 12 m (Mean ± SD)
Test	Panwar et al., 2021 [25]	3.200 ± 0.4216	2.950 ± 0.2838	NR	−0.250 ± 0.2635	NR
Control	Panwar et al., 2021 [25]	2.8 ± 0.788	3.400 ± 0.864	NR	0.65 ± 0.411	NR
Test	Lee et al., 2023 [27]	4.83 ± 1.32	NR	NR	NR	−0.80 ± 1.11
Control	Lee et al., 2023 [27]	4.70 ± 1.16	NR	NR	NR	−0.18 ± 1.07
Test	Abbas et al., 2020 [28]	4.786 ± 1.286	NR	4.429 ± 1.058	NR	NR
Control	Abbas et al., 2020 [28]	4.429 ± 1.397	NR	3.571 ± 1.397	NR	NR

KMW: keratinized tissue width; SD: standard deviation; NR: not reported; FU: follow-up; m: months.

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
