# Peer review of "Immediate Implant Placement with Soft Tissue Augmentation Using Acellular Dermal Matrix Versus Connective Tissue Graft: A Systematic Review and Meta-Analysis"

_materials, 2024, doi:10.3390/ma17215285_

Round 1
Reviewer 1 Report
Comments and Suggestions for Authors
This systematic review and meta-analysis aims to investigate the efficacy of connective tissue grafting versus acellular dermal matrix for soft tissue management in immediate implant placement.
My comments and suggestions for revising the manuscript are as follows:
Please pay attention and prepare your manuscript according to MDPI style (abstract without headings, figures and tables inserted in the text, number and use italics, not underlined, for subsections, references style, authors contribution etc.)
Please rephrase the problematic parts of your manuscript, the plagiarism check reported a 40% match, which is quite high, even for a review type paper.
Once you gave the abbreviation in brackets, please use the abbreviation only, do not repeat the full term followed by the abbreviation and or use the full term.
Watch out for the typing errors.
As I said before, please insert the figures and tables in the text, the text should have a title and figures should have a caption, so the Supplementary Materials part, containing the legends, should be removed, as well as Appendix B.
Appendix A should be mentioned in the text, similar to figures and tables.
Comments on the Quality of English Languagenone
Author Response
Dear reviewer, Thanks for your comments. Here is a table explaining the suggested changes and the changes made. All these modifications are marked in red in the manuscript.
|
CHANGES TO MAKE |
CHANGES MADE |
|
Please pay attention and prepare your manuscript according to MDPI style (abstract without headings, figures and tables inserted in the text, number and use italics, not underlined, for subsections, references style, authors contribution etc.)
|
Changes and marked with red |
|
Please rephrase the problematic parts of your manuscript, the plagiarism check reported a 40% match, which is quite high, even for a review type paper.
|
We send it again to our English language editing. The 40% match due to primary sources 1, 2, 3, 6 presenting 7%, 4%, 3%, 1% respecting which is taken from the template of the PRISMA checklist located on page 11- 16 on the manuscript.
|
|
Once you gave the abbreviation in brackets, please use the abbreviation only, do not repeat the full term followed by the abbreviation and or use the full term.
|
Changes and marked with red |
|
Watch out for the typing errors.
|
We send it again to our English language editing and corrections made in Red |
|
As I said before, please insert the figures and tables in the text, the text should have a title and figures should have a caption, so the Supplementary Materials part, containing the legends, should be removed, as well as Appendix B. Appendix A should be mentioned in the text, similar to figures and tables.
|
Changes and marked with red |

Reviewer 2 Report
Comments and Suggestions for Authors
The present article investigates the efficacy of connective tissue grafting 11 (CTG) versus acellular dermal matrix (ADM) for soft tissue management in immediate implant 12 placement (IIP). The study focuses on comparing soft tissue thickness (STT) and keratinized tissue width (KTW) changes post-implantation.
- The manuscript is clear, relevant for the field and presented in a well-structured manner. The design is pertinent.
- The satistics of the study is well performed with relevant results.
- The manuscript is scientifically sound.
- The manuscript’s results are reproducible.
- Please include in the introduction more recent published studies.
- The conclusion is relevant.
- Also please perform a revion of english.
- The article is very important for the research and presents the latests publications in the fild.
- The statistics is very well performed using an apropriate soft for the design of the statistics.
Moderate
Author Response
Dear reviewer, Thanks for your comments. Here is a table explaining the suggested changes and the changes made. All these modifications are marked in red in the manuscript.
|
CHANGES TO MAKE |
CHANGES MADE |
|
Please include in the introduction more recent published studies.
|
Corrections made in red |
|
Also please perform a revion of english.
|
We send it again to our English language editing and corrections made in red |

Reviewer 3 Report
Comments and Suggestions for Authors
The study presents a a systematic review on the implant placement with soft tissue augmentation using acellular dermal matrix versus connective tissue graft. As a characteristic of any review, the study doesn't add to the treatment procedures but makes a reasoning over existing published studies. The document well mentions the limitations of the study, namely problems of heterogeneneity that have to be considered, limited number of studies and needed follow-up to the presented study. The review was conducted in according with PRISMA and STROBE which seem adequate and for that adding value to the present review. In general, even with the presented limitations, the conclusions add to the existing knowledge thus the study seems worth to publish.
Author Response
Dear reviewer,
Thank you for considering this article.
Reviewer 4 Report
Comments and Suggestions for Authors
-In the introduction, the authors should consider integrating findings from recent literature on soft- and hard-tissue healing, such as the article : https://www.sciencedirect.com/science/article/pii/S030057122400263X?via%3Dihub on healing dynamics. This study provides crucial insights into tissue regeneration, which could help contextualize the comparison between ADM and CTG in enhancing peri-implant soft tissue outcomes.
-The introduction would also benefit from discussing the broader comparison of grafted versus non-grafted IIP. Previous studies suggest that soft tissue augmentation improves esthetics and reduces marginal bone loss in grafted IIP, compared to non-grafted cases, which often show inferior soft tissue thickness and keratinized tissue width (KTW).
-However, the study lacks a forest plot comparing these techniques directly, which is a significant oversight given the title's focus.
- Future studies should adopt a well-designed randomized controlled trial (RCT) protocol to directly compare ADM and CTG in IIP. Below is a suggested points of the protocol:
- Study Design:
- Population:
- Inclusion Criteria:
- Exclusion Criteria:
- Intervention:
- Outcome Measures:
- Assessment Methods:
- Statistical Analysis:
- Follow-up Period:
Author Response
Dear reviewer, Thanks for your comments. Here is a table explaining the suggested changes and the changes made. All these modifications are marked in red in the manuscript.
|
CHANGES TO MAKE |
CHANGES MADE |
|
-In the introduction, the authors should consider integrating findings from recent literature on soft- and hard-tissue healing, such as the article : https://www.sciencedirect.com/science/article/pii/S030057122400263X?via%3Dihub on healing dynamics. This study provides crucial insights into tissue regeneration, which could help contextualize the comparison between ADM and CTG in enhancing peri-implant soft tissue outcomes.
|
Corrections made in red, however the article provided is about Diabetes and Peri-implantitis, so we think that it makes no sense to introduced to the article because is not the topic to be discussed. |
|
-The introduction would also benefit from discussing the broader comparison of grafted versus non-grafted IIP. Previous studies suggest that soft tissue augmentation improves esthetics and reduces marginal bone loss in grafted IIP, compared to non-grafted cases, which often show inferior soft tissue thickness and keratinized tissue width (KTW).
|
Corrections made in red in the introduction |
|
-However, the study lacks a forest plot comparing these techniques directly, which is a significant oversight given the title's focus.
|
We believe that figures 2-4 and 6 are sufficient. |

Reviewer 5 Report
Comments and Suggestions for Authors
This systematic review evaluated the effectiveness of connective tissue grafting compared to acellular dermal matrix for managing soft tissue in immediate implant placement. The paper adhered to PRISMA guidelines and evaluated for changes in STT and KTW following grafting procedures in IIP. The authors concluded that no statistically significant differences between CTG and ADM in STT gains at the 12-month mark, with CTG showing a mean difference.
Overall, This paper is in good quality. Both CTG and ADM in this paper were proofed to be viable options for soft tissue management in immediate implant placement, with no significant differences in terms of STT and KTW outcomes, suggesting the need for more standardized research in the long-term. I would like to support this manuscript if the authors can solve the following issues.
1. Line 50-60 should be simplifed. This paragraph doesn't show tight connection with the introduction.
2. Table 2-7 should be in scientifc format.
Author Response
Dear reviewer, Thanks for your comments. Here is a table explaining the suggested changes and the changes made. All these modifications are marked in red in the manuscript.
|
CHANGES TO MAKE |
CHANGES MADE |
|
Line 50-60 should be simplifed. This paragraph doesn't show tight connection with the introduction. |
Changed: Funato et al. describes and classifies the 4 scenarios of which immediate implants can be considered. However, he explains that the extraction sockets with four walls show better aesthetic predictability [7]. Whereas Buser et al. describes protocols according to the time of placement. The theory is that in the type 1 protocol, being extraction, with immediate implant placement directly into the alveolar socket via flapless implant placement with osseous augmentation and a connective tissue or allograft, being a predictable approach [8].
|
|
Table 2-7 should be in scientifc format. |
Changed and market in red |
